# Overview of Solid Lipid Nanoparticles in Breast Cancer Therapy

**DOI:** 10.3390/pharmaceutics15082065

**Published:** 2023-07-31

**Authors:** Kyumin Mo, Ayoung Kim, Soohyun Choe, Miyoung Shin, Hyunho Yoon

**Affiliations:** 1Department of Medical and Biological Sciences, The Catholic University of Korea, Bucheon 14662, Republic of Korea; mo.gyumin8521@gmail.com (K.M.); hydroxyl.dragon@gmail.com (A.K.); soohchoe16@gmail.com (S.C.); 2Department of Biotechnology, The Catholic University of Korea, Bucheon 14662, Republic of Korea; 3Department of Pathology, Yale University School of Medicine, New Haven, CT 06510, USA; miyoung.shin@yale.edu

**Keywords:** solid lipid nanoparticle, breast cancer, drug delivery, cancer therapy

## Abstract

Lipid nanoparticles (LNPs), composed of ionized lipids, helper lipids, and cholesterol, provide general therapeutic effects by facilitating intracellular transport and avoiding endosomal compartments. LNP-based drug delivery has great potential for the development of novel gene therapies and effective vaccines. Solid lipid nanoparticles (SLNs) are derived from physiologically acceptable lipid components and remain robust at body temperature, thereby providing high structural stability and biocompatibility. By enhancing drug delivery through blood vessels, SLNs have been used to improve the efficacy of cancer treatments. Breast cancer, the most common malignancy in women, has a declining mortality rate but remains incurable. Recently, as an anticancer drug delivery system, SLNs have been widely used in breast cancer, improving the therapeutic efficacy of drugs. In this review, we discuss the latest advances of SLNs for breast cancer treatment and their potential in clinical use.

## 1. Introduction

Lipid nanoparticles (LNPs) are nonviral delivery systems for small-molecule therapeutics. LNPs are generally composed of four lipids, including phospholipids, especially 2-Distearoyl-sn-glycero-3-phosphocholine (DSPC) or 1,2-Dioleoyl-sn-Glycero-3 Phosphatidylethanolamine (DOPE). They are less than 100 nm in diameter [1]. LNPs are formed by adding ionizable or cationic lipids to liposomes to encapsulate negatively charged oligonucleotides via electrostatic reactions [2]. A crucial component of the delivery system is the ionizable lipid, which is an amphiphilic molecule with a cationic head group and two hydrophobic tails. Uncharged LNPs commonly bind to the cell surface via hydrophobic interactions or endocytosis. When LNPs are internalized into a cell, the acidic state of the endosome promotes the cationization of ionizable lipids, destroying the endosome membrane and releasing RNA into the cytoplasm [3]. LNPs have mainly been studied for mRNA delivery in vaccines. In addition, LNPs have recently been developed not only as vaccine platforms but also for cancer and gene therapies. According to a recent study, LNPs can be functionalized with antibodies for anticancer drug treatment. Especially, antibody-functionalized liposomes are able to target angiogenesis, uncontrolled cell proliferation, and tumor cells with lipid-based nanoparticles. Most of them have been in phase 1 clinical trials [4].

Liposomes, lipospheres, and microsimulation carrier systems have defects such as rapid drug release, instability, and oxidation. The use of liposomes as drug delivery systems, including indomethacin, amphotericin B, and azidothymidine, have been made available to the commercial market to date [5]. In the 1990s, a new drug delivery system called solid lipid nanoparticles (SLNs), which are generally spherical and 50–1000 nm in diameter, was developed. SLNs overcome various limitations by combining several of the beneficial features of polymeric nanoparticles. The advantages of SLNs include low toxicity, efficient drug targeting, regulated drug release, high drug loading, and prevention from degradation [6]. SLNs are carrier systems that have aqueous colloidal dispersions and a solid biodegradable lipid matrix for efficacious water-soluble treatment, which provides impressive properties, including small size, a broad surface area, and high medication stacking. In detail, in A549 cell lines, SFN-loaded SLNs at concentrations of 500 and 700 μg/mL, lowered the cell viability to 62% and 46% (depending on the SFN concentrations of, respectively, 12.1 and 17.7 μM). Using free SFN at identical concentrations, the cell viability was dropped to 68% and 79%, respectively [7]. These features are attractive owing to their potential to improve the performance of pharmaceuticals. SLNs are a new potential colloidal transporter technique and an alternative to polymers, which are distinct from oil-in-water emulsions [8]. Furthermore, lipid-based nanosystems, more than any other nanomaterials, have the potential to act as alternatives for the development of the posterior segment of eye target products [9]. The primary aim of SLNs in drug delivery is to improve the bioavailability and efficacy of agents and to regulate the nonspecific toxicity and immunogenicity of drugs, especially poorly soluble drugs [10]. It can also be utilized in cancer therapy with these advantages.

Nowadays, many researchers grapple with developing better treatments for numerous cancers in multifaceted ways. Breast, lung, and colorectal cancers account for 51% of all new diagnoses in women, with breast cancer accounting for nearly one-third of all cases [11,12]. Due to improved treatments and mammographic screening techniques, breast cancer diagnosis rates are steadily increasing, while mortality rates are decreasing [13]. However, breast cancer remains difficult to treat and is responsible for the deaths of many women, and also occurs in men. The heterogeneity of breast cancer, including its distinct phenotypes and morphologies, results in different clinical behaviors that necessitate the development of context-specific therapy approaches. Triple-negative breast cancer (TNBC) is the most aggressive type of breast cancer. Chemotherapies such as carboplatin or cisplatin are widely used for TNBC, but the therapeutic effect of these therapies is not significant [14,15]. As one of the supplementary methods with the capacity to overcome the limitations of conventional therapy, SLNs have become attractive as a means to target breast cancers. There are many trials for breast cancer treatment using LNPs. For instance, RNA interference-mediated treatment has been drawn attention as a treatment for TNBC. LNP-encapsulating siRNA was developed with an antibody, and inhibited tumor growth in mouse model [16], indicating that LNPs and SLNs are considered as a promising therapy for cancers. Hereby, this review highlights the therapeutic advances in the use of SLNs for breast cancer.

## 2. Role of SLNs in Delivery Systems

### 2.1. SLNs

SLNs are composed of a lipid matrix, surfactants, and sometimes co-surfactants at physiological temperatures [17]. SLNs can be formed through various methods, including high-shear homogenization and ultrasound, hot homogenization, cold homogenization, strong-pressure homogenization, solvent emulsification, evaporation, and microemulsion [18]. A microemulsion is an easy method that does not require high-level equipment or high-energy input and uses an organic solvent [19]. In addition, there are some drug incorporation models for SLNs, such as the solid solution and core-shell models [20]. In the drug-enriched shell model, when the recrystallization temperature of the lipid is reached, a solid lipid core is generated, and the drug is concentrated in the SLN liquid outer shell by decreasing the dispersion temperature. Cooling the nanoemulsion in the drug-rich core model results in the saturation of the dissolved drug in the molten lipid at saturation solubility. After precipitation of the drug occurs before recrystallization of the lipid, recrystallization of the lipid surrounding the drug occurs because of extreme cooling [21].

SLNs are suitable for intravenous applications by uncomplicated dispersion in solutions to treat various chronic diseases, such as diabetes, obesity, neurodegenerative diseases, and cancer [22]. It has been reported that SLNs can be easily internalized and can be proposed as a surrogate colloidal drug delivery system for drug administration, especially for malignant melanoma and colorectal cancer, suggesting that SLNs influence antitumor activity [10]. Compared to other nanoparticles, SLNs present significant features. SLNs allow for the possibility of scaled-up production [23], decrease biotoxicity while enhance bioavailability [24,25], and can be used in versatile applications with hydrophilic and lipophilic drugs [26]. Furthermore, engineered SLNs can effectively deliver DNA to the binding sites, resulting in poor transfection efficacy and cytotoxicity. Thus, SLNs cover all sites for drug delivery and can be administered via transdermal penetration, oral absorption, and injection, suggesting that SLNs have potential as carriers of bioactive materials (Figure 1). Although SLNs have been introduced as attractive LNPs, there are still some limitations to overcome, such as expulsion and the low loading efficiency of the drug, which has led to the development of the next generation of nanoparticles and nanostructured lipid carriers (NLCs) [27,28]. NLCs facilitate high drug solubility, have an increased loading capacity, and increased storage duration [29] (Table 1). Hence, further studies need to be conducted for optimal lipid nanoparticles.

### 2.2. SLNs as Drug Delivery Systems

To provide more efficient treatment, delivery techniques based on the advantages of SLNs have been developed. To overcome the limitations of conventional technologies, SLNs have attracted attention because of their unique technological and biological properties [37]. Consequently, SLNs have emerged as feasible nanocarriers for drug delivery that complement polymeric nanoparticles [38]. Drug delivery systems encompass formulations, approaches, and technologies for specific purposes [39]. The goal of successful drug delivery is to reduce problems such as side effects, non-specificity, burst or interrupted release, and toxicity [26,40]. The biodegradability and biocompatibility of SLNs are their most well-known characteristics [10,41]. SLN biodegradability and biocompatibility, which are based on the nature of the lipids, provide diverse possibilities for administration, and lower the risk of safety issues.

#### 2.2.1. Combination of SLNs with Hydrophilic Compounds

SLNs can be combined with hydrophilic and hydrophobic drugs to stabilize and protect the drugs from harsh conditions such as degradation [8,37]. As the original delivery efficiency of drugs, lipids, and hydrophilic compounds is poor, the use of SLNs can offer improved drug delivery mechanisms [40]. It can contribute to broad applications with drugs of various natures due to the unique benefits of SLNs. Owing to the solid lipid matrix, hydrophobic molecules can be encapsulated using appropriate surfactants [41]. Hydrophilic compounds interact insufficiently with lipophilic molecules and divide into aqueous phases, resulting in poor drug delivery and release [42]. To overcome this problem, the double emulsification method is widely used to carry hydrophilic drugs. Hydrophilic drugs liquefy in water and lipid materials melt in an organic solvent to circumvent drug partitioning. A primary water-in-oil (w/o) emulsion is formed using a dispersed water phase in lipid solvents. A double emulsion, water-oil-water (w/o/w), is obtained when the primary emulsion is emulsified with an aqueous surfactant and the lipid solvent is evaporated [43,44]. However, there is a possibility of toxicity because of the residue from organic solvents, which can be solved using the melt dispersion technique in combination with the double emulsion method [42]. Consequently, bioavailability is enhanced via absorption from various sources of the drug.

#### 2.2.2. Versatile Administration

SLN-based technologies can increase bioavailability, particularly when administered orally [45]. Generally, intravenous injection is the main administration route of chemotherapy. It has the advantages of dose control and cytotoxicity; however, it can cause side effects, such as venous thrombosis and normal cell attack [24]. In contrast, the use of SLNs offers improved bioavailability as well as multi-route administration, particularly via the oral route [46]. Using LNPs, numerous drugs can overcome low oral bioavailability resulting from a lack of absorption [47]. Chemical and enzymatic barriers to drug passage through the gastrointestinal (GI) tract make it difficult for some drugs to function properly because of poor permeability, instability, and poor water solubility [45]. Nevertheless, the larger surface areas of insoluble compounds with smaller particle sizes improve resistance to extreme conditions in the GI tract and increase the efficacy of drug absorption [47]. For instance, the oral bioavailability of simvastatin, which is known to be 5%, can be improved with SLNs according to one study [48]. Furthermore, parenteral administration can be enhanced by overcoming the above-mentioned issues. Particularly in anticancer treatment, drug delivery using SLNs can be favorable for patients in terms of convenience and the potential for self-administration [19]. As such, SLNs provide a wide range of administration routes, including non-parenteral and parenteral routes.

Nanotechnology attenuates toxicity, enables different routes of administration, and controls drug delivery and release [37]. It also specifies the target tissue, allows large-scale production in cost-effective ways, and co-delivery of more effective therapeutic strategies [37]. The advantages of SLNs offer pivotal capabilities, especially in drug delivery systems. Thus, enhanced drug delivery systems have been focused on the use of SLNs to achieve improved therapeutic effects.

### 2.3. Nucleic Acid Delivery Systems

Since the COVID-19 pandemic, mRNA vaccines have attracted the attention of many researchers. Accordingly, methods to improve the efficacy of gene therapies, such as mRNA and DNA vaccines, which refers to the delivery system, have followed. Moreover, a codelivery mechanism of SLNs related to small interfering RNA (siRNA) and microRNAs (miRNAs) offers a trailblazing cancer treatment that addresses drug resistance [19]. The advantages of SLNs, including controlled and targeted drug release, reduced side effects, and improved bioavailability, are related to improved nucleic acid delivery. Nucleic acid delivery is key in terms of efficiency and safety.

#### 2.3.1. mRNA Vaccines

To bind negatively charged nucleic acids, SLNs must be oppositely charged. This charge is determined by both cationic lipid and solid lipid-cationic lipid interactions that can cooperate well with cationic lipids and simultaneously retain a solid structure for stability [49]. The electrostatic interaction between SLNs and nucleic acids causes encapsulation of nucleic acids, preventing enzymatic degradation and promoting mobility [49]. Moreover, mRNA is a meaningful non-viral transfection vector because of its unnecessary access to the genome, resulting in a decreased risk of mutation and cancer [50]. mRNA-based treatment is considered a promising therapy for several applications, such as vaccines, anticancer treatment, and genome editing [51]. It also renders safer and fast expression, prolonged stability, and higher reproducibility, all of which are beneficial in regard to vaccine production [49,50].

mRNA vaccine delivery vehicles are mainly based on LNPs for the successful transportation and protection of mRNA [52]. LNP-based mRNA vaccination is currently used by BioNTech/Pfizer and Moderna [53]. Furthermore, an in vivo study conducted in TNBC cells presents significant synergistic functions of paclitaxel and p53-mRNA [54]. It implied that nanocarriers are considered innovative adjuvants in vaccine delivery systems in terms of encapsulation protection from the host milieu and long-term immunostimulation [54]. They are also expected to be useful in anticancer therapy. The entrapment of mRNA with nanocarriers induces endosomal escape and cellular uptake, as well as mRNA delivery into immune cells, which can promote both innate and adaptive immune responses [55]. Using mRNA with LNPs can deliver encoded cytokines for a reinforced immune response and improve safety by targeting desired cells with less systemic toxicity [56].

#### 2.3.2. DNA Vaccine

DNA vaccines have been introduced as prospective methods for inducing both cellular and humoral immunological responses [57]. For various diseases, including cancer, DNA vaccines with SLN-induced targeted therapies are based on immune responses [58]. Reducing the side effects of a vaccine, while sustaining its desired effects, is one of the remaining issues that researchers are currently addressing. To maintain transfection longer, the combination of DNA vaccines with SLNs shows a noticeable balance between containment and the regulated release of nucleic acids, which is crucial [49]. SLNs act as gene carriers, enhancing the stability and delivery of DNA vaccines, and are able to protect drugs from degradable environments [28]. Unlike RNA, DNA must pass through the nuclear membrane, requiring different delivery strategies for RNA and DNA [49]. Conjugated with surface ligands, LNPs can easily bind to, and recognize, target cells and enhance their cellular uptake [59]. Moreover, cationic SLNs interact with negatively charged DNA at higher concentrations in a safer way [60]. LNP formulations have developed thresholds by improving stability, overcoming physiological barriers, and the possibility of targeted therapy, similar to drugs [51]. To enhance the benefits of SLNs and create more refined therapeutic delivery mechanisms, codelivery of drugs and genes has been developed [19]. The efficacy of antitumor treatments has been improved through the use of SLNs that combine plasmid DNA and anticancer drugs [49]. Advanced nanodelivery technology offers novel adjuvant benefits for medical platforms.

## 3. Antitumor Effect of SLNs in Breast Cancer

### 3.1. Enhancing Bioavailability of Anti-Breast Cancer Drugs

Bioavailability refers to the rate of drug absorption into the bloodstream so that its desired effect becomes available. Thus, high bioavailability is an essential component of antitumor drug efficacy as it is strongly related to the method of administration and drug toxicity [61]. Oral administration is easy, convenient, low in cost, and has proven efficacy [61]. Hydrophilic drugs can be administered orally; however, hydrophobic, or amphiphilic drugs have limited routes of administration [62]. The use of SLNs has emerged as an effective method for improving drug bioavailability and enabling oral intake of hydrophobic and amphiphilic drugs [47]. In addition to their increased solubility, SLNs exhibit many other remarkable properties such as small particle size, high zeta potential, and chemical and mechanical stability. When antitumor drugs are encapsulated in SLNs, the physiological lipid core prevents drug degradation and improves the stability of SLN-loaded drugs [63]. Zeta potential is a major parameter that indicates the electrostatic charge on the surface of particles dispersed in a liquid medium [8]. The high zeta potential of LNPs, either positive or negative, means that the particles have sufficient stability for aggregation. These advantages make SLNs drug delivery systems promising supplementary methods for enhancing the bioavailability of antitumor drugs.

Cisplatin, a chemotherapeutic drug, is widely used to treat several types of cancers, particularly TNBC. Cisplatin is a DNA-inserting agent that cross-links DNA, inducing interference in RNA transcription and DNA replication; thus, it exerts antitumor effects by suppressing cell proliferation and promoting programmed cell death [64]. However, the administration of cisplatin is limited owing to its side effects, including renal failure, ototoxicity, loss of hearing, cerebral blindness, tubular necrosis, and papilledema. Recently, SLN encapsulation was found to be an ideal method for overcoming the low bioavailability of cisplatin [65]. The cisplatin-SLN-treated MCF-7 cells effectively inhibited cell proliferation. Moreover, cisplatin-SLNs showed powerful cytotoxic effects in MCF-7 cells 6.51 ± 0.39 µg/mL of IC50 value compared to free cisplatin with 10 µg/mL of IC50 value, but not in normal cells [66]. These results demonstrate that cisplatin-loaded SLNs have overcome the toxicity and limitations of administration methods in patients with breast cancer.

Moreover, SLNs are beneficial for the bioavailability of antitumor molecules from natural sources. Although natural components with antitumor effects are less toxic, they are limited by their low solubility and bioavailability. Wenrui Wang et al., showed that curcumin-loaded SLNs (Cur-SLNs) are more biocompatible than natural curcumin [67]. Curcumin is a natural hydrophobic polyphenol molecule that has been used as a drug against cancer, viral infections, and inflammation because of its antitumor activity [68]. In breast cancer therapy, curcumin enhances antitumor activity by targeting cell cycle mechanisms, p53-dependent apoptosis, PI3K/Akt/mTOR signaling, NF-κB transcription factors, and tumor angiogenesis [69]. Many in vivo studies have shown that curcumin inhibits metastasis and proliferation of breast cancer cells [70]. Despite its promising anti-cancer effects, the clinical application of curcumin has some limitations owing to its hydrophobicity, which results in quick elimination and instability in the body [71]. An evaluation showed that the Cur-SLNs have sufficient zeta potential to repel each other, which is helpful for enhancing their stability and drug bioavailability [67].

### 3.2. Overcoming Multidrug Resistance (MDR)

Although many antitumor drugs for breast cancer have been developed commercially, drug resistance remains a major challenge. MDR lowers drug efficacy through various mechanisms that lead to the ineffective action of drugs on cancer cells and failure to respond to treatment among patients [72]. Offering multiple benefits, including lower toxicity and improved stability of the drug, SLNs can overcome the MDR of antitumor drugs [19]. Recently, SLNs have opened new avenues for resolving the MDR problem associated with various anti-breast cancer drugs.

Tamoxifen is known to be an effective drug for the endocrine therapy of HR-positive breast cancer, especially for ER-positive subtypes. Tamoxifen is part of the endocrine therapy class, which is a selective estrogen receptor modulator that inhibits the transcriptional activity of the ER [73]. Although tamoxifen has been used for the past 40 years, approximately 50% of patients with ER-positive breast cancer are not cured because of tamoxifen resistance [73]. Tamoxifen is an ER antagonist that exerts therapeutic effects by mediating ER signaling. Thus, the major cause of antiestrogen resistance is the loss of ER expression and function in cancer cells [74]. Additionally, the ATP-binding cassette (ABC) transporter family, a drug efflux transporter, is involved in chemoresistance mechanisms in breast cancer [75]. The expression of ABC family proteins increases in tamoxifen-resistant breast cancer cells [76]. Long-term administration of tamoxifen induces resistance in breast cancer cells via these pathways. Malfunction of ABC family mechanisms includes the overexpression of permeability glycoprotein (P-gp) as a result of the malfunctioning expression of the ABC transporter family [77]. Tamoxifen-SLNs reverse P-gp-mediated tamoxifen resistance by inducing apoptosis [77]. The apoptotic effect of SLNs has the potential to overcome MDR during breast cancer treatment. Tamoxifen-loaded SLNs have the potential to induce apoptosis and accumulate cells, even in tamoxifen-resistant cells, which can reverse drug resistance acquired in the G0/G1 phase [77,78]. Tamoxifen-SLN treatment suppressed the expression of anti-apoptotic genes and miRNAs (miR-497 and miR-1280) were also suppressed in tamoxifen-SLN treatment [78]. Therefore, the apoptotic effect of SLNs is a promising approach for increasing drug efficacy and reversing drug resistance in breast cancer (Figure 2).

### 3.3. Role of Ligand-Loaded SLNs in Breast Cancer Therapy

Most side effects of antitumor drugs are caused by damage to normal cells because of the low specificity of the drugs. To overcome this limitation, various methods have been developed to increase the specificity of a drug by conjugating it with a tumor-specific antibody or ligand. SLNs conjugated with hormone receptor-specific antibodies enhance the targeting potential for breast cancer cells. Compared to SLN treatment alone, some in vitro and in vivo studies have shown enhanced antitumor effects of cationic SLNs linked to CAB51 and HER2 binding antibodies [79]. The linkage between SLNs and CAB51 can be achieved via streptavidin-biotin interactions. Incubation of the SLN-streptavidin complex with the biotinylated antibody mixture forms the SLN-streptavidin-antibody [79]. As the zeta potential is closely related to physical stability, these methods are affected by changes in the zeta potential of the nanoparticle surface [80]. According to studies on the treatment of HER2-overexpressing cells with normal SLNs compared to antibody-conjugated SLNs, a greater reduction in cancer cell viability was observed with antibody-conjugated sentinel node treatment [79]. In addition, treatment with the antibody-SLN complex showed much lower cancer cell viability than CAB51 treatment alone at the same dose [79]. These synergistic effects of SLNs linked to the HR antibody proved promising for specific antitumor effects.

However, HR-targeting treatments are not effective for HR-negative breast cancer. Unlike the HR-positive subtype, TNBC can be targeted via death receptor-5 (DR5) because TNBC cells have a higher proportion of DR5 on the cell surface than normal cells [81]. Recent studies have shown that DR5 antibody-conjugated SNPs have high cytotoxicity and decreased off-target effects in TNBC cells, both in vitro and in vivo [82]. These effects seem to result from the synergistic activity between SLNs and DR5 antibodies, which leads to site-specific delivery [82]. In addition to antibody-conjugated SLNs, SLNs have been modified with cancer cell-targeting ligands such as folic acid (FA) to increase drug specificity. FA-conjugated SLNs combined with antitumor drugs, such as letrozole (LTZ), enhance drug efficacy by inducing cancer-specific apoptosis in breast cancer [83]. These results suggest that ligand-loaded SLNs are a promising approach to target-specific breast cancer treatment. With this concept, more patients are expected to overcome these diseases without experiencing side effects.

## 4. Clinical Implications of SLNs in Breast Cancer Therapy

In breast cancer treatment, chemotherapy is the preferred method using anticancer drugs, including doxorubicin (DOX), docetaxel (DTX), and paclitaxel (PTX) [84]. DOX is an anthracycline antibiotic that is used as a cytostatic agent. However, DOX induces toxic side effects and leads to hair loss. In addition, MDR causes elevated transport of drugs out of tumor cells, resulting in cancer therapy failure [85]. On the other hand, the arginine-glycine-aspartic (RGD) tripeptide is used for neovasculature delivery with high binding efficacy. Various RGD-binding nanoparticles have been developed for targeted drug delivery to breast cancer [86].

DOX-loaded SLNs (RGD-DOX-SLNs) encapsulated DOX in SLNs via modified RGD and pH sensitivity. Their pharmacokinetics and tissue distribution leads to long-term circulation and high stability of RGD-DOX-SLNs to accomplish effective accumulation in tumor tissues. Treatments with DOX-SLNs showed increased suppression of tumor growth compared to treatment with DOX alone. The accumulation of nanoparticles in the tumor region was enhanced. In addition, efficient encapsulation and stimulated intracellular release of DOX may lead to improved antitumor efficacy. Intravenously administered RGD-DOX-SLNs showed significantly more effective anti-tumor effects than DOX-SLNs, in which the RGD was not modified. This demonstrates that modification of the ligand induced a critical benefit for SLNs. Hence, RGD-DOX-SLNs are a promising lipid carrier system for improving the treatment of breast cancer [87].

LTZ is one of the most effective treatments for hormone-dependent breast cancer [88]. As LTZ carriers, SLNs can overcome systemic toxicity through nonspecific drug targeting [89]. FA receptor-mediated endocytosis of FA-SLNs-LTZ was attributed to increased caspase-3 activity and DNA fragmentation in MCF-7 cells. Furthermore, in vivo experiments evaluating the level of tissue caspase-3 showed remarkably elevated apoptosis induction after treatment, suggesting that FA-SLNs-LTZ may be useful for caspase-3-mediated apoptosis in a target-specific manner [83].

DTX is an FDA-approved agent that is generally used to treat different types of cancers, including breast, ovarian, and prostate cancers. DTX is a cytostatic agent that regulates the growth of tumor tissue and induces cell-cycle arrest by reversibly attaching to microtubules and inducing temporary structural stabilization [90]. In particular, DTX has shown improved survival associated with metastatic disease compared to other chemotherapeutic drugs; however, clinical intravenous administration of DTX is limited by its low water solubility [91]. SLN-DTX was stable and had a high encapsulation efficiency of approximately 86%, resulting in high cytotoxicity in cancer cells by arresting cell cycle progression in the G2/M phase. SLN-DTX treatment inhibits 4T1 breast cancer metastasis to the lungs by reducing the number and size of tumor nodules. Interleukin-6 (IL-6) is a cytokine that plays a role in controlling pro-inflammatory and metastatic tumors. Previous studies have shown that circulating IL-6 levels increased in breast cancer, resulting in increased cancer progression and metastasis [92,93]. Tumor-bearing mice expressed high levels of IL-6 and developed lung and liver metastasis. An in vivo study using immunohistochemical analysis showed that SLN-DTX significantly inhibited tumor growth in 4T1 breast cancer cells, prevented lung metastasis at a dose of 10 mg/kg, and did not cause significant systemic toxicity in mice [94]. These results indicate that SLN-DTX could be an alternative and potential carrier for breast cancer treatment and metastasis prevention.

Previous studies have shown that the tumor suppressor gene p53 is highly altered in patients with TNBC. A novel treatment strategy was developed using a combination of chemotherapeutic agents and tumor suppressor p53 mRNA. PTX combined with camptothecin (CPT) has several advantages over monotherapy. The co-delivery of PTX and CPT has better treatment efficacy for cancer therapy compared to monotherapy [95]. PTX and CPT couple to ionizable amino lipids to form polysaccharide amino acid lipids (PAL). PAL lipids were formulated with *p53* mRNA and incorporated in the LNPs. PAL-p53 LNPs intravenously injected into TNBC nude mice showed notably higher antitumor effects than free drugs [96]. Therefore, LNP-mRNA-mediated treatment in combination with chemotherapy can offer enormous synergistic effects in TNBC treatment (Table 2).

## 5. Conclusions

Although the survival rate of breast cancer patients continues to improve with the development of efficient diagnosis and treatment strategies, breast cancer still accounts for a significant portion of cancer-related deaths in women. In addition, the characteristics of conventional anticancer drugs, such as high toxicity, low specificity, and MDR, must be addressed to improve the efficacy of drugs in breast cancer. SLNs can overcome these obstacles by improving the bioavailability, stability, and interaction with cancer-specific peptides, suggesting significant potential for breast cancer treatment. Many studies have confirmed the enhanced efficiency of SLN-loaded anti-breast cancer drugs both in vitro and in vivo. However, although SNL-mediated drug delivery systems have great potential for clinical application, there have been insufficient clinical trials of SLNs for breast cancer treatment due to the relatively new system. Thus, further research should focus on preclinical and clinical trials using advanced SLNs for practical application in breast cancer patients.

## Figures and Tables

**Figure 1 pharmaceutics-15-02065-f001:**
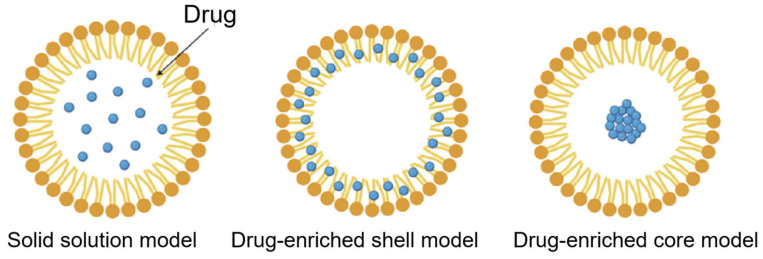
Schematic drug loading models of SLNs; solid solution model, drug-enriched shell model, and drug-enriched core model. The solid solution model mainly consists of a dispersed drug in the solid lipid matrix and also has a high lipophilic interaction. In a drug-enriched shell model, the encapsulated drug is located in the outer shell and conversely, the drug-enriched core model shows the concentrated drug in the core. These drug distribution models are distinguished by their preparation methods and can be chosen for specific purposes in terms of size, characteristics, and release of the drug.

**Figure 2 pharmaceutics-15-02065-f002:**
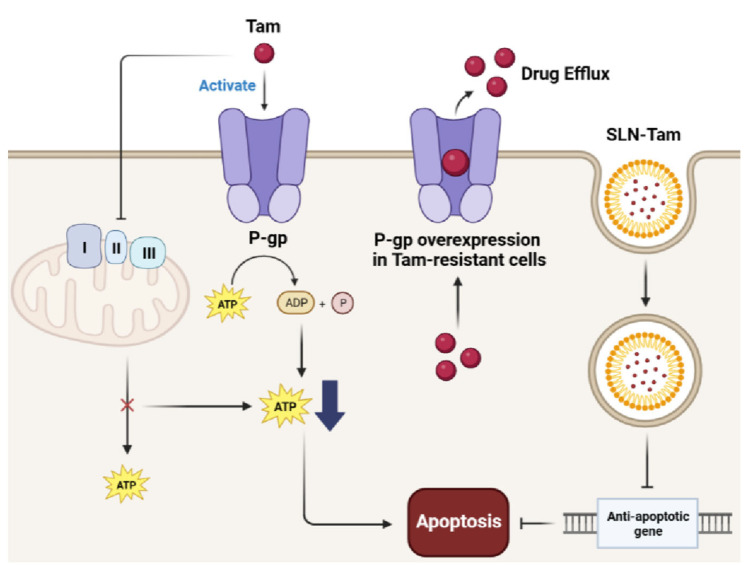
Role of SLN-tamoxifen in tamoxifen-resistant cancer. Tamoxifen induces apoptosis of cancer cells via activating P-glycoprotein (P-gp) and inhibiting complex I-III in mitochondria. Complex I–III are mitochondria electron transport complex mediating ATP synthesis. However, P-gp is also known as a representative drug efflux transporter. In tamoxifen-resistant cancer cells, P-gp overexpression enhances drug efflux activity rather than the antitumor effect. SLN-tamoxifen overcomes resistance by delivering drugs directly to cytosol and inhibiting expression of anti-apoptotic genes.

**Table 1 pharmaceutics-15-02065-t001:** Advantages and disadvantages of SLNs.

SLN Type	Study Type	Advantages	Disadvantages	Critical Remark	References
Solid solution model	In vitro	- Long-term drug release capacity - High correlation with the lipid	- Drug stability issues under certain conditions	Efficient when combined with lipophilic drugs	[21,30,31,32]
Drug-enriched shell model	In vitro	- Effective burst release - Risen drug solubility	- Inappropriate for extended release of active pharmaceutical ingredients	Necessity of controlled burst release	[32,33,34,35]
Drug-enriched core model	In vitro	- Regulated drug release	- Lack of drug loading ability	Application with highly concentrated drugs	[19,32,34,36]

**Table 2 pharmaceutics-15-02065-t002:** Clinical implications of drugs loaded SLNs/LNP in breast cancer therapy.

Name	Effect	Loaded Drug	Clinical	References
Res-SLNs	- Inhibit cell proliferation in dose-dependent in MDA-MB-231 cell line - Enhance cell cycle arrest in the G0/G1 phase through Cyclin D1 downregulation	Resveratrol	In vitro	[97]
RGD-DOX-SLNs	- Suppress tumor growth - Enhance antitumor effects after intravenously administration	Doxorubicin	In vitro, In vivo	[98,99]
sFA-SLNs-LTZ	- Overcome systemic toxicity - Increase caspase-3 activity, leading to apoptosis	Letrozole	In vitro, In vivo	[83,89]
SLN-DTX	- Arrest G2/M phase - Inhibit breast cancer metastasis and tumor growth	Docetaxel	In vitro, In vivo	[94,100]
PAL-p53-LNP	- Enhance antitumor effect after intravenous administration	Paclitaxel	In vitro, In vivo	[54,101]
FA-DATS-SLNs	- Increase cytotoxicity in aggressive TNBC - Upregulate pro-apoptotic caspase-9 while downregulating anti-apoptotic proteins (Bcl2)	Folic acid conjugated diallyl trisulfide	In vitro	[102]
CS/Lf/PTS-SLNs	- Show significant antitumor activity in TMBC - Predominently reduce tumor growth and promote apoptotic, antiangiogenic activity in orthotopic breast cancer model	Chondroitin/Lactoferrin- dual functionalized pterostilbene	In vitro, In vivo	[103]

## Data Availability

Not applicable.

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
