# Peer review of "Overview of Solid Lipid Nanoparticles in Breast Cancer Therapy"

_pharmaceutics, 2023, doi:10.3390/pharmaceutics15082065_

Round 1

Reviewer 1 Report

The manuscript focuses on the “Overview of Solid Lipid Nanoparticles in Breast Cancer Therapy”. The topic is interesting and up-to-date in the Pharmaceutics field. However, there are several recente reviews addressing the topic, and the present manuscript does not give much more new contribution. More critical is that the manuscript is poorly organized and presente several issues:

-The abstract is poor and not focused in breast cancer. It should better highlight the contente of the manuscript.

-The paper may have some gender equality issues when the authors describe that “However, breast cancer remains difficult to treat and is responsible for the deaths of many women.”.It is well known that breast cancer also has a relevant incidence in men, and that has to be described as well.

-The organization of the sections is poor and with little sense. For instance section 2.2 is “SLNs as drug delivery systems”, and section “2.2.1 Combination of SLNs with hydrophilic compounds”. Why just to focus on hydrophilic compounds when SLN are mostly used to deliver hydrophobic compounds due to its nature? More relevant why just focus on oral delivery in section 2.2.2. when most of the exemples given in the manuscript are focused in parental delivery.

-Another issue is the section 2.3. Nucleic acid delivery systems mostly focused in mRNA and DNA vacines. It is not clearly described and thus not understandable its relation with breast cancer.

A More critical scientific error is the section 3.1. Bioavailability of SLNs in anti-breast cancer drugs. The concept of bioavailability concerns to drugs and not to carriers, so the concept of bioavailability of SLN is completely wrong.

The conclusion section is very poor, and it should not contain references. Even more strange is providing a table of the advantages and disadvantages of SLN in the conclusion,when should be placed at the beggining of the manuscript.

Minor editing of English language required. A review should be performed to correct some spell and grammar issues.

Reviewer 2 Report

The manuscript by Mo et al., describe about the lipid nanoparticles and its use in breast cancer therapy. The review article focused on different aspect of the lipid nanoparticles like role of lipid nanoparticles in drug delivery, its antitumor effect on breast cancer, and its clinical implications. The manuscript is very well described and focused on all the aspects of lipid nanoparticles.

Round 2

Reviewer 1 Report

The authors had the humbleness to follow the reviewers advice and improved the manuscript quality. It may be considered for publication now.